# Measuring Interprofessional Collaboration’s Impact on Healthcare Services Using the Quadruple Aim Framework: A Protocol Paper

**DOI:** 10.3390/ijerph20095704

**Published:** 2023-05-01

**Authors:** Yang Yann Foo, Xiaohui Xin, Jai Rao, Nigel C. K. Tan, Qianhui Cheng, Elaine Lum, Hwee Kuan Ong, Sok Mui Lim, Kirsty J. Freeman, Kevin Tan

**Affiliations:** 1Department of Technology Enhanced Learning and Innovation, Duke-NUS Medical School, Singapore 169857, Singapore; 2Health Services Research Unit, Singapore General Hospital, Singapore 169608, Singapore; 3Department of Neurosurgery, National Neuroscience Institute, Singapore 308433, Singapore; 4Duke-NUS Medical School, Singapore 169857, Singapore; 5Department of Neurology, National Neuroscience Institute, Singapore 308433, Singapore; 6Department of Neuroradiology, National Neuroscience Institute, Singapore 308433, Singapore; 7Health Services & Systems Research, Duke–NUS Medical School, Singapore 169857, Singapore; 8Department of Physiotherapy, Singapore General Hospital, Singapore 169608, Singapore; 9Singapore Institute of Technology, Singapore 138683, Singapore; 10Office of Education, Duke–NUS Medical School, Singapore 169857, Singapore

**Keywords:** Interprofessional Collaboration (IPC), return on investment (ROI), Quadruple Aim, patient outcomes, patient experience, provider wellbeing, cost of care, multimethod approach, time-driven activity-based costing (TDABC)

## Abstract

Despite decades of research on the impact of interprofessional collaboration (IPC), we still lack definitive proof that team-based care can lead to a tangible effect on healthcare outcomes. Without return on investment (ROI) evidence, healthcare leaders cannot justifiably throw their weight behind IPC, and the institutional push for healthcare manpower reforms crucial for facilitating IPC will remain variable and fragmentary. The lack of proof for the ROI of IPC is likely due to a lack of a unifying conceptual framework and the over-reliance on the single-method study design. To address the gaps, this paper describes a protocol which uses as a framework the Quadruple Aim which examines the ROI of IPC using four dimensions: patient outcomes, patient experience, provider well-being, and cost of care. A multimethod approach is proposed whereby patient outcomes are measured using quantitative methods, and patient experience and provider well-being are assessed using qualitative methods. Healthcare costs will be calculated using the time-driven activity-based costing methodology. The study is set in a Singapore-based national and regional center that takes care of patients with neurological issues.

## 1. Introduction

Despite decades of research on the impact of interprofessional collaboration (IPC), we still lack definitive proof that team-based care can lead to tangible effect on healthcare outcomes [1,2]. In an age of evidence-based healthcare, the uncertainty around IPC’s return on investment (ROI) means that institutional support for IPC will remain variable and fragmentary; healthcare leaders and policymakers cannot justifiably throw their weight behind massive manpower reforms that are needed to facilitate IPC [3,4]. 

The lack of definitive proof for the ROI of IPC is likely due to two reasons: lack of a unifying conceptual framework with which to measure the healthcare outcomes of IPC, and over-reliance on the single-method study design for examining a phenomenon as complex and multi-faceted as IPC.

### 1.1. Conceptual Framework and Operationalization Issues

Notwithstanding IPC’s purported goal to improve patient care, IPC literature does not commonly report on its contribution to achieving the goals of healthcare, namely population health, patient experience, and cost of care, commonly known as Triple Aim [5]. In a systematic review [6] of interprofessional collaborative practices, only 25 or 18.8% of all the 133 studies reviewed reported one or two healthcare outcomes of the Triple Aim. Of these 25, 22 focused on patient experience of care alone and only 3 examined patient outcomes. None of the studies investigated the cost of IPC. The more recent Cochrane systematic review of randomized controlled trials of IPC interventions [2] shows that the situation has not changed. Out of the nine studies reviewed by Reeves and colleagues, only two measured IPC’s impact on healthcare outcomes, while the remaining seven examined more distal outcomes such as healthcare professionals’ (HCP) adherence to recommended practice, use of healthcare resources, team communication, and coordination. Whether and how these distal outcomes contributed to the healthcare system were unknown. Thus, to strengthen the evidence base for the ROI of IPC, it will be helpful to investigate IPC’s effectiveness using a unifying conceptual framework that encompasses all important components of healthcare outcomes.

A strong candidate is the Quadruple Aim [7], an expansion of the Triple Aim [5], where a fourth outcome—provider well-being—was added to the original framework. Although the Quadruple Aim framework was designed as a system of metrics for population health, the term population is not limited to people residing in a specific geographic area. The essence of this concept is its specificity and comprehensiveness, so that all the four outcomes can be measured in an integrated manner. In the context of IPC, population refers to patients with a specific disease receiving care from a specific IPC practice or practices in a specific healthcare institution and population health refers to patient outcomes. In short, the Quadruple Aim potentially can become a unifying framework with which researchers could identify outcomes for investigating IPC’s effectiveness holistically, and cumulatively this may help to strengthen the evidence base for IPC’s ROI.

### 1.2. Complex Problem Requiring Multimethod Approach

The other reason why IPC still lacks definitive proof of ROI could arguably be traced to the over-reliant use of a single-method (usually quantitative) study design [8]. While quantitative studies are useful, there are also limitations. Non-randomized quantitative studies have collectively produced a small body of evidence that consistently showed IPC to be effective in improving well-measured patient outcomes such as reducing fall risks [9], lowering blood pressure [10,11,12,13] and HbA1c in patients with diabetes [14,15]. However, when quantitative studies examined outcomes that involved social and behavioral dimensions, IPC was shown to have little or conflicting effects. A cluster RCT involving 236 hospital patients (intervention = 144 cases) found no significant differences in patients’ satisfaction with care and could not explain why [16]. Another large case control study involving more than 300 thousand primary care patients found that team-based primary care was associated with substantial reduction in emergency department and hospital utilization among chronically ill patients, but among healthier patients hospitalization and outpatient utilization actually increased [17]. The authors of this study speculated that coordinated care may have resulted in higher outpatient utilization among healthier patients but could not determine why hospitalization among this group also increased [17]. Such studies looking at one single aspect of the ROI of IPC are prone to generate mixed evidence which makes it difficult for healthcare leaders and policymakers to rely on them to inform their decisions on the investment of IPC.

Thus, the need for an approach that leverages both quantitative and qualitative methods in a complementary manner to evaluate IPC’s contribution to achieving the Quadruple Aim emerges. Quantitative methods are useful for measuring patient outcomes and cost of care as they are quantifiable. But for patient satisfaction and provider well-being, both of which involve interactions that are dynamic, variable, and non-linear [18], qualitative research is required for understanding how IPC works, for which patients and providers, in what context, and why [19].

We would like to propose an approach that may be new or unfamiliar to IPC researchers. Extant literature shows that the impact of IPC on the cost of care is often measured by shorter length of stay or lower rate of readmission [17,20,21]. However, such calculation methods have several limitations. The first study [17] stated that their primary source of data were administrative claims and not detailed clinical information retrieved from electronic health records, and there was erroneous attribution of patients to physicians, and physicians to practices. In the second study [20], the researchers were unable to use the calculation methods to determine which IPC process intervention had the greatest impact. In the last study of patients with chronic pancreatitis [21], the researchers omitted to track the cost associated with patients utilizing the services of an additional behavioral and psychiatric treatment team.

In view of the above limitations, this study presents time-driven activity-based costing (TDABC) [22] as an alternative research methodology. Originating from the field of management studies, TDABC has been recognized as the most accurate and efficient method of calculating the direct cost of complex team-based care to date [23]. Adopting the TDABC method to calculate the cost of IPC will thus be an important step towards accurately estimating the ROI of IPC.

### 1.3. Aims of Research Protocol

This study’s protocol aims to illustrate the feasibility of using the Quadruple Aim as a framework to measure IPC’s ROI by adopting a multimethod approach (Figure 1). The protocol describes two studies with several inter-related aims (see Supplementary Material: Appendix A. Proposal Workplan). Study 1 assesses IPC’s impact on patient outcomes and calculates the cost of care using quantitative methods. Study 2 seeks to understand the impact of team-based care resulting from IPC on patient experience and provider well-being using qualitative methods. Findings from these two inter-related studies are integrated to provide multi-faceted insights on IPC’s ROI, which may serve to guide organizations and policymakers on issues such as whether, how, and when healthcare budgets should be invested /disinvested in IPC.

## 2. Methods

This protocol paper describes two studies using the four dimensions of the Quadruple Aim framework to assess the impact of IPC on healthcare outcomes. The sections entitled ‘Study Setting’ and ‘Selection of the IPC Teams to Study’ will apply to both studies. However, the information for each study will be provided in a separate section to enhance readability.

### 2.1. Study Setting

The study setting is the National Neuroscience Institute (NNI) in Singapore, a national and regional center for treating the conditions of the brain, spine, nerves, and muscles. NNI operates out of two main hospitals (Tan Tock Seng Hospital and Singapore General Hospital) and four partner hospitals (Changi General Hospital, KK Women’s’ and Children’s Hospital, Khoo Teck Puat Hospital, and Seng Kang General Hospital) in Singapore. Specialist physicians, nurses, and allied health professionals (AHP) namely radiographers, therapists, psychologists, and medical technologists from the Departments of Neurology, Neurosurgery, Neuroradiology, Neurodiagnostic, and Neuromuscular Laboratories integrate their expertise and work together to care for patients in both inpatient and outpatient settings in these hospitals. Inpatient care is delivered by multidisciplinary teams of physicians, nurses, and AHP. These teams consist of a combination of core fixed members (physicians of varying seniority; four to six members) and fluid members (nurses and AHP of the wards). Outpatient care is mainly delivered by fixed teams of nurses and physicians (two to four members).

### 2.2. Selection of IPC Teams to Study

We will use a purposive sampling strategy known as positive deviance sampling, which allows researchers to compare individuals or communities who have discovered solutions to problems (positive deviants) with those peer individuals or communities who continue to struggle with the problem [24]. In this study, we define positive deviant as a group of HCP who manage to find solutions to work as an integrated team with common goals despite operating in an environment characterized by individualized episodic work practices.

Positive deviance sampling strategy is useful for facilitating culture change because it identifies examples that have found solutions to problems and difficulties within a given organization itself, rather than impose change from external sources [25,26]. Examples of positive deviance include three outpatient teams that care for patients with specific disease conditions: Persistent Concussion Clinic (PCC), Neuroimmunology Clinic (NC), and Epilepsy Clinic (EC) [27]. If the results of the proposed study show that the team-based approach produces better patient outcomes, provides care that patients need, and improves HCP well-being at a lower cost, then it could serve as a persuasive example for other services within the institution to adopt similar IPC practices.

### 2.3. Study 1 Design: Assessing IPC’s Impact on Patient Outcomes and Cost of Care

#### 2.3.1. Design

We will use a retrospective cohort study design for the assessment of patient outcomes and TDABC methodology for the assessment of cost of care.

#### 2.3.2. Study Sample

The population for population health in this study refers to patients with neurological disorders receiving care from the three selected team-based practices in NNI. For the assessment of patient outcomes, all patients who have been receiving care in the three selected team-based clinics since the start of these clinics (2021 for PCC, 2013 for NC, 2014 for EC) and have at least one-year follow-up period as of July 2022 will be included in the study sample.

The calculation of the cost of care does not involve sampling. We will survey all HCP involved in delivering care in the three selected clinics at the time of the study on the average amount of time they spend on each of the activities specified in a care pathway.

#### 2.3.3. Measurements and Data Collection Plan

For patients receiving care in the PCC for head injuries which are acute, the primary patient outcome measure will be time to symptom resolution. The symptoms include vertigo, convergence insufficiency, cognitive functioning, psychological functioning, sleep disorders, and headache. For patients with epilepsy or neuroimmunological disorders which are chronic, the primary outcome measures include both disease activity and patient adherence to medication. Disease activity for epilepsy is measured by seizure frequency, and that for neuroimmunological disorders is measured by clinical (relapses) or radiological (MRI scan abnormalities) activity after initiation of immunotherapy. Medication adherence is based on patient self-reporting and is documented during each clinic visit. Secondary outcomes are process outcomes directly related to IPC for each clinic: rate of identification of symptoms by AHP for PCC, completion rate of screening for medication adherence, time to initiation of immunotherapy for NC, and tolerance and complications for EC. In addition to outcome measures, we will also collect data on patients’ demographic and clinical characteristics at the baseline (i.e., when they were first referred to these clinics), and treatment regimen during the study period. All data will be extracted from electronic medical records upon appropriate institutional approval.

Following the TDABC methodology [22], calculation of the cost of care will involve specification of essential care activities along each care pathway, estimation of time spent on each activity by each group of HCP, and ascertainment of the unit manpower cost for each group of HCP. We will first map the care pathway consisting of all essential care activities in sequential order for each of the three selected clinics following the best practice of process mapping [28]. Training will be provided to the whole study team by one of the study team members (EL) who has experience in process mapping. The training serves to equip the study team with necessary knowledge of the importance and procedure of this step. We will also engage an expert in clinical operations from the Support Operations and Strategic Management Analytics Department in NNI to co-develop these care pathways. After the care pathways are drawn up, we will survey the HCP performing each activity on each pathway to estimate the average amount of time required to carry out the specific activity using a pre-designed form. If more than one individual from the same HCP group is involved in a specific activity, the average of their estimates will be used as the time taken for that group of HCP for that specific activity. The unit cost data for each group of HCP will be obtained from the finance department of the institution.

#### 2.3.4. Analysis Plan

Survival analysis will be used to characterize the primary outcome of time to symptom resolution for patients in PCC and the process outcome of time to initiation of immunotherapy for patients in NC. Cox proportional hazards regression analysis will be performed to examine these time-to-event outcomes after adjusting for the baseline demographic, clinical characteristics, and the treatment regimen during the follow-up period. Generalized Linear Mixed Models (GLMM) will be used to analyze the longitudinal primary outcomes of patients in NC and EC from the shortest period of one year to the longest period of eight years adjusting for baseline characteristics and treatment regimen during the follow-up period. Other process outcome results will be obtained through simple arithmetic calculations.

We will calculate the cost of each activity along the care pathway by multiplying the time spent on each activity by a specific HCP and the unit cost for the HCP performing the activity. The cost of an episode of care will be the sum of the costs of all activities on the care pathway. The concept of ‘episode of care’ in this protocol will be determined by the care pathway of the respective clinics. The total cost for each clinic will be calculated by multiplying the cost of each episode of care by the volume of patients treated during the study period.

### 2.4. Study 2 Design: Understanding How IPC Impacts Patient Experience and Provider Well-Being

#### 2.4.1. Sampling and Recruitment

For the patient experience component of Study 2, all patients who have been receiving care in the three selected team-based clinics since the start of these clinics (2021 for PCC, 2013 for NC, 2014 for EC) and have at least a one-year follow-up period as of July 2022 will meet the inclusion criterion. We will be using a purposive sampling strategy known as team sampling unit (TSU), where we interview [29] the patients as well as the HCP they identify as being vital to their care. The interviews will be conducted separately. This sampling strategy has been successfully used to produce rich analyses by researchers studying team-based care for patients with severe chronic obstructive pulmonary disease [30,31] and advanced heart failure [32,33]. As TSU has the potential of offering more opportunities for data triangulation than individual interviews, it seems particularly relevant for our study as it will enable us to collect multiperspective information to derive a more complete understanding of whether and how PCC, NC, and EC provide team-based care that meets patients’ needs. TSU is recommended to consist of the patient and at least two HCPs [31]. Thus, we will also plan our recruitment strategy in a similar fashion. 

We will also seek the patients’ consent to shadow them [34,35]. We will explain that shadowing will allow us to better understand how they actually experience care in the PCC, NC, and EC settings. However, should patients only consent to be interviewed and not be shadowed or identify the HCP, we will still include them in the study as it is often challenging to recruit patients. For patients who agree to identify the HCP, the latter will be subsequently voluntarily recruited. Both patients and HCP will be interviewed at a time and location of their choosing.

Patient recruitment will be done by one of the research team members (QC) who is a neutral party not involved in the care of the patients. To avoid any impression of coercion, the consent form will state that, regardless of their decision to participate in the study, patients will continue to receive the same level and quality of care as before. As patient gender, age, and severity of condition may influence the team’s attitudes and actions regarding care [33], patients will be recruited based on these characteristics. In cases where the patient is mentally incapable of providing consent, we will recruit the caregiver as a proxy participant.

Study 2 also aims to examine the impact of IPC on provider well-being. As PCC, NC, and EC comprise only between two and four individuals—the specialist physician, the advanced practice nurse or nurse clinician, a physiotherapist, and an occupational therapist—we will use as our sampling a strategy known as complete target population [24]. As suggested by the name, we will voluntarily recruit and interview everyone in this unique group of interest. By doing so, we hope that we would be able to gather information that will help us understand whether and how working as a team to care for patients affects HCPs on issues such as their perceived well-being (and its inverse, namely burnout and stress), job satisfaction, work engagement, and intention to leave [8].

We anticipate that some of the HCP to be recruited for this part of the study may not be identified as part of their patients’ TSU. Even so, if the HCP give consent we will interview and observe them because the small size of the PCC, NC, and EC teams does not provide us with viable alternatives. Furthermore, it is challenging to get clinicians to consent to be interviewed due to the time-consuming nature of qualitative data collection. Hence, it is prudent to engage any HCP who agree to participate in the study.

We plan to also collect observation data because while semi-structured interviews have the potential of allowing us to unveil the HCP’s personal experiences, this mode of data collection may not always reflect the actual interactions and hence observations might serve as a useful complementary qualitative data collection tool [24]. However, observations give rise to concerns about participant reactivity [36]. To avoid affecting the regular interactions among the HCP, non-participant observation methods will be used: we will observe at a distance where participants’ speech can be heard clearly, but without being so close as to be intrusive or cause disruption.

Recruitment for both components of Study 2 will continue until we reach data sufficiency, which is when we are able to construct findings that elucidate in a rich and nuanced way the impact of IPC on patient experience and provider well-being. Should the response be poor, we will seek the help of PCC, NC, and EC nurses to encourage the patients by explaining how their participation may help to improve care delivery. To avoid a conflict of interest, the nurses will not engage in direct recruitment.

#### 2.4.2. Data Collection Plan

A team of trained qualitative researchers (YYF, XX, QC) with no prior relationships with or influence over the patients, their caregivers, and HCP will be involved in the data collection. The three researchers will be rostered based on availability to conduct semi-structured interviews [33]. We will iteratively collect and analyze data and refine the topic guide to help us access the most illuminating information. The semi-structured interviews will be audio-recorded with participant consent and transcribed verbatim. Observation notes will be made [37] focusing on patient interactions and communication with HCP, as well as how providers interact within the team and respond to each other’s suggestions and resolve disagreements. For the patients, the researchers will also shadow them [34,35] and make observation notes describing how patient care is provided at the PCC, NC, and EC outpatient clinics. 

#### 2.4.3. Analysis Plan

De-identified interview transcripts and observation notes will be inductively analyzed using reflexive thematic analysis (RTA) [38,39]. Using RTA’s six phase approach, investigators YYF and XX will begin by familiarizing themselves with the data. This will be followed by phase two where codes will be constructed at both semantic (descriptive) and latent (interpretative) levels. The codes will then be developed into candidate themes in the third phase and reviewed in the fourth to ensure that the early themes are coherently supported by the verbatim quotes. In the fifth phase, the reviewed themes will be finalized and defined. The sixth and last phase involves the writing up of the report. The entire research team will discuss the content of codes and themes through a process known as thematic mapping [39].

### 2.5. Effect of the COVID-19 Pandemic on Data Collection

The threat of the pandemic on healthcare systems globally may be receding but occasional surges in cases within hospital settings mean that researchers may be unable to collect data in person. In view of such contingencies, we are prepared to conduct our interviews with patients, their caregivers, and HCP over Zoom communications. This is provided that the participants have access to the internet, Zoom software, and possess the requisite technological know-how. For patients or caregivers who lack technical affordances, we will postpone the collection of data until such a time when it is deemed safe for research to resume. Patient shadowing and observations of healthcare teams will similarly be postponed. 

## 3. Conclusions

This research protocol aims to address the seemingly intractable problem of measuring IPC’s ROI. Without addressing this critical gap, evidence for the ROI of IPC will remain fragmented which makes it difficult for healthcare leaders and policymakers to appreciate the value of IPC objectively. Hence, we seek to demonstrate the feasibility of measuring IPC’s ROI by operationalizing the Quadruple Aim framework using a multimethod approach. Specifically, quantitative methods are used to measure patient outcomes and cost of care, and qualitative methods to understand how team-based care impacts patient experience and provider well-being.

These aims notwithstanding, this protocol has several limitations. For study 1, the evaluation of patient outcomes adopted a single-arm design due to the unavailability of a parallel comparison group in NNI and the unavailability of electronic medical records of a historical comparison group. As a result, we are unable to make a statistical comparison of the patient outcomes of team-based care with those of usual care. Instead, a non-statistical comparison with the treatment outcome for patients with the same diseases reported in the published literature will be made to interpret the findings of this part of the study. However, as this protocol is meant to test the feasibility of our concept of how to measure IPC’s ROI, future studies could prospectively plan to collect data for a comparison study.

As for Study 2, interview and observation data have inherent limitations. While semi-structured interviews may help researchers gather hard to access information such as perceptions and experiences of patients and HCP, the data may be limited by recall bias or not reflect actual interactional patterns. Observation as a mode of data collection may also raise concerns about participant reactivity, where participants may act differently because of the presence of a researcher. For these reasons, study 2 is designed to collect both interview and observation data for triangulation purposes. 

Projecting ahead, if the two empirical studies described in this protocol prove to be effective in measuring the ROI of IPC, researchers may proceed to examine other important issues on whether, how, and when healthcare budgets should be invested/disinvested in IPC. From the perspective of practical import, this will be helpful in guiding healthcare leaders and policymakers to make evidence-based decisions in the provision of healthcare. From the theoretical perspective, being able to strengthen the evidence base of IPC’s ROI will also help to advance this field of research.

## Figures and Tables

**Figure 1 ijerph-20-05704-f001:**
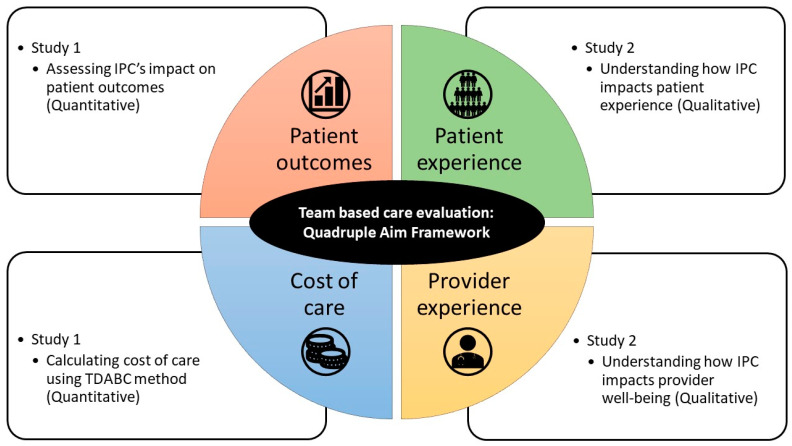
Multimethod approach to study IPC’s ROI using Quadruple Aim as framework.

## Data Availability

Data availability statement not applicable as this is a protocol paper.

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
