# Peer review of "Measuring Interprofessional Collaboration’s Impact on Healthcare Services Using the Quadruple Aim Framework: A Protocol Paper"

_ijerph, 2023, doi:10.3390/ijerph20095704_

Round 1
Reviewer 1 Report
I believe this is an important study that was thoughtfully proposed. While I rate the article as scholarly, I did struggle with two items and have two suggestions.
One is the definition of population health. As stated in the article: "...the term population is not limited to people residing in a specific geographic area. The essence of this concept is its specificity and comprehensiveness, so that all the four outcomes can be measured in an integrated manner. In the context of IPC, population refers to the patients who are cared for by HCP engaged in a form of IPC and population health refers to patient outcomes." I see where the authors are coming from but I wonder whether the population might be defined as persons with neurological disorders?
I recognize this is a "Protocol Paper" but I would like there to have been at least a test of this protocol even in one or two participants. The authors noted the challenges they could face in collecting the data. I would like to have seen some reference to their testing the protocol in any way so they could speak from experience about the potential challenges.
I liked the figure but I feel like the table is redundant.
This is a complex study with a number of unique methodologies and analyses. I don't know if it is possible but having a visual of all the parts, maybe in the form of a flowchart, might make it easier to see how the study would flow in real time.
Author Response
Response to Reviewer 1 Comments
Point 1: I believe this is an important study that was thoughtfully proposed. While I rate the article as scholarly, I did struggle with two items and have two suggestions.
One is the definition of population health. As stated in the article: "...the term population is not limited to people residing in a specific geographic area. The essence of this concept is its specificity and comprehensiveness, so that all the four outcomes can be measured in an integrated manner. In the context of IPC, population refers to the patients who are cared for by HCP engaged in a form of IPC and population health refers to patient outcomes." I see where the authors are coming from but I wonder whether the population might be defined as persons with neurological disorders?
Response 1: Please provide your response for Point 1.
We thank Reviewer 1 for her/his comments.
The reviewer’s interpretation of population for our proposed study as patients with neurological disorders is correct. We have added a statement “The population for population health in this study refers to patients with neurological disorders receiving care from the three selected team-based practices.” in the Methods Section (lines 185-186). We have also revised the statement in lines 73-74 to make it clearer that population in the context of IPC refers to patients with a specific disease receiving care from a specific IPC practice or practices in a specific healthcare institution.
Point 2: I recognize this is a "Protocol Paper" but I would like there to have been at least a test of this protocol even in one or two participants. The authors noted the challenges they could face in collecting the data. I would like to have seen some reference to their testing the protocol in any way so they could speak from experience about the potential challenges.
Response 2: Please provide your response for Point 2. (in red)
We agree with Reviewer 1’s allusion to the fact that protocol papers do not typically contain pilot data as they are descriptions of methods for conducting empirical studies (e.g., Anderson et al., 2020). However, we have duly noted her/his concerns about the challenges highlighted in the protocol paper, specifically challenges in recruiting healthcare professionals (HCP) and patients. We would like to assure Reviewer 1 that we have experience in addressing these concerns (Foo et al., 2022; Chew et al., 2019). Hence we seek Reviewer 1’s understanding for not conducting a pilot.
Point 3: I liked the figure but I feel like the table is redundant.
This is a complex study with a number of unique methodologies and analyses. I don't know if it is possible but having a visual of all the parts, maybe in the form of a flowchart, might make it easier to see how the study would flow in real time.
Response 3: Please provide your response for Point 2. (in red)
We have removed the table.
To make it easier how the proposed studies would be carried out, we have added a visual called ‘Protocol Workplan’ in the Supplementary Material section.
References
Anderson, J. E; Aase. K.; Bal, R., et al. (2020). Multilevel influences on resilient healthcare in six countries: an international comparative study protocol. BMJ Open, 10(12), e039158. DOI: 10.1136/bmjopen-2020-039158
Chew, LC, Xin, X., et al. (2019). An evaluation of the Virtual Monitoring Clinic, a novel nurse-led Service for monitoring patients with stable rheumatoid arthritis. International Journal of Rheumatic Diseases, 22(4): 619–25. https://doi.org/10.1111/1756-185X.13436.
Foo, Y.Y.; Tan, K., Rao, J., et al. (2022). Viewing interprofessional collaboration through the lens of networked ecological systems theory. Journal of Interprofessional Care, 36(6), 777-785. DOI: 10.1080/13561820.2021.2007864

Reviewer 2 Report
The protocol is well thought out; however, the manuscript provides no preliminary pilot data. In addition, some of the language is more subjective than objective, and the use of first person throughout the manuscript must be removed.
I recommend doing a small pilot test and then resubmitting with some preliminary findings.
Here are examples on how to change language to make it more objective:
Line 2: Correct spelling of Fulfill, it needs a second “l” at the end of the word.
Line 35: Not sure about including Quadruple Aim as keyword. It’s a bit jargony.
Line 98: Don’t use first person in journal articles. Suggest changing “we” to the authors
Lines 98-100: The opening sentence of the paragraph is editorializing. Make it objective vs. subjective. For example:
Thus, the need for an approach that leverages both quantitative and qualitative methods in a complementary manner to evaluate IPC’s contribution to achieving the Quadruple Aim emerges.
Lines 102-103: Change useful to required. Suggested revision. “…qualitative research methodologies are required for understanding…”
Line 105: Delete the first sentence (On the outcome of care…), because this paragraph is not about your proposed approach but about the limitations of the approaches that have been used.
Line 110: “Imperfect” is an awkward word choice; better word is erroneous.
Line 116: Remove first person (we). Instead say something like, In view of the above limitations, this study presents time-driven activity-based costing (TDABC) as an alternative research methodology.
Line 123: Remove first person (our). This study’s proposed protocol…
Line 129: Change “can be” to “are.”
Figure 1 and Table 1: Change “Multimethod” to Mixed-methods”
Lines 136-139: Suggested edit to remove first person and shift to past tense because the study was already conducted:
To assess the impact of IPC on healthcare outcomes, two studies were evaluated using the four dimensions of the Quadruple Aim framework. The sections entitled ‘Study Setting’ and ‘Selection of the IPC Teams to Study’ apply to both studies. The information for each study is provided in a separate section to enhance readability.
Line 142: Remove first person and change to past tense. Change “Our” to “The” and “is” to “was”—The study setting was the…
Line 157: Change to past tense. Change “will use” to “used.” We used purposive sampling…
Line 160: Change to past tense. Change “we define” to “was defined.” In this study, a positive deviant was defined as…
Lines 166-167: Remove first person (our, we). Change to “Examples of positive deviants include… [27].”
Line 169: Remove first person. Change “our” to “the.”
Author Response
Response to Reviewer 2 Comments
The protocol is well thought out; however, the manuscript provides no preliminary pilot data. In addition, some of the language is more subjective than objective, and the use of first person throughout the manuscript must be removed.
Point 1: I recommend doing a small pilot test and then resubmitting with some preliminary findings.
Response 1: Please provide your response for Point 1. (in red)
We thank Reviewer 2 for her/his comments.
We have duly noted Reviewer 2’s recommendation that we do a small pilot. However, protocol papers are descriptions of methods for conducting empirical studies and do not include pilot data (e.g., Anderson et al., 2020). Reviewer 2 may have concerns regarding the feasibility of the empirical studies described in the protocol, specifically in the areas of recruiting health care professionals (HCP) and patients. We would like to assure Reviewer 2 that we have experience in overcoming these challenges (Foo et al., 2022; Chew et al., 2019). Hence we seek Reviewer 2’s understanding for not conducting a pilot.
Point 2: Here are examples on how to change language to make it more objective:
Response 2: Please provide your response for Point 2. (in red)
|
Reviewer 2’s comment |
Authors’ response |
|
Line 2: Correct spelling of Fulfill, it needs a second “l” at the end of the word. |
Changes have been made |
|
Line 35: Not sure about including Quadruple Aim as keyword. It’s a bit jargony.
|
Quadruple Aim is a term well-understood in both IPC and healthcare research in general. Furthermore, it is the conceptual framework underpinning our protocol paper. It is thus important that we include it as a keyword.
|
|
Lines 98-100: The opening sentence of the paragraph is editorializing. Make it objective vs. subjective. For example:
Thus, the need for an approach that leverages both quantitative and qualitative methods in a complementary manner to evaluate IPC’s contribution to achieving the Quadruple Aim emerges.
|
Changes have been made |
|
Lines 102-103: Change useful to required. Suggested revision. “…qualitative research methodologies are required for understanding…” |
Changes have been made |
|
Line 105: Delete the first sentence (On the outcome of care…), because this paragraph is not about your proposed approach but about the limitations of the approaches that have been used. |
Changes have been made |
|
Line 110: “Imperfect” is an awkward word choice; better word is erroneous. |
Changes have been made |
|
Line 116: Remove first person (we). Instead say something like, In view of the above limitations, this study presents time-driven activity-based costing (TDABC) as an alternative research methodology. |
Changes have been made |
|
Line 123: Remove first person (our). This study’s proposed protocol… |
Changes have been made |
|
Line 129: Change “can be” to “are.” |
Changes have been made |
|
Figure 1 and Table 1: Change “Multimethod” to Mixed-methods” |
We thank Reviewer 2 for the suggestion but would like to clarify that we are not using the mixed-methods approach, which involves the collection, analysis and integration of ‘both’ quantitative and qualitative data (Creswell, 2015). However, our aim is different: we hope to use multiple forms of quantitative and qualitative data (Creswell, 2015) to assess the various dimensions of the Quadruple Aim and triangulate their respective outcomes to derive a holistic understanding of interprofessional collaboration’s return on investment. As such, we will retain the use of the term, ‘multimethod’.
|
|
Lines 136-139: Suggested edit to remove first person and shift to past tense because the study was already conducted:
To assess the impact of IPC on healthcare outcomes, two studies were evaluated using the four dimensions of the Quadruple Aim framework. The sections entitled ‘Study Setting’ and ‘Selection of the IPC Teams to Study’ apply to both studies. The information for each study is provided in a separate section to enhance readability. |
We have changed ‘Our protocol’ to ‘This protocol’. However, as we have not conducted these two studies, we will retain the use of the present and future tenses in this paragraph. |
|
Line 142: Remove first person and change to past tense. Change “Our” to “The” and “is” to “was”—The study setting was the… |
We have replaced first person, ‘Our’, with the definite article, ‘The’. However, as we have not conducted the two empirical studies proposed in this protocol, we will retain the present tense used in this sentence. |
|
Line 157: Change to past tense. Change “will use” to “used.” We used purposive sampling… |
As we have not conducted the two empirical studies proposed in this protocol, we will retain the future tense used in this sentence. |
|
Line 160: Change to past tense. Change “we define” to “was defined.” In this study, a positive deviant was defined as… |
As we have not conducted the two empirical studies proposed in this protocol, we will use the present tense in this sentence. |
|
Lines 166-167: Remove first person (our, we). Change to “Examples of positive deviants include… [27].” |
Changes have been made |
|
Line 169: Remove first person. Change “our” to “the.” |
Changes have been made |
References
Anderson, J. E; Aase. K.; Bal, R., et al. (2020). Multilevel influences on resilient healthcare in six countries: an international comparative study protocol. BMJ Open, 10(12), e039158. DOI: 10.1136/bmjopen-2020-039158
Chew, LC, Xin, X., et al. (2019). An evaluation of the Virtual Monitoring Clinic, a novel nurse-led Service for monitoring patients with stable rheumatoid arthritis. International Journal of Rheumatic Diseases, 22(4): 619–25. https://doi.org/10.1111/1756-185X.13436.
Creswell, J.W. (2015). A Concise Introduction to Mixed Methods Research. Sage.
Foo, Y.Y.; Tan, K., Rao, J., et al. (2022). Viewing interprofessional collaboration through the lens of networked ecological systems theory. Journal of Interprofessional Care, 36(6), 777-785. DOI: 10.1080/13561820.2021.2007864

Reviewer 3 Report
Dear Author,
The goal of this study protocol is clearly described.
The method is clear and appropriate.
However, there is a lack of information on the Singapore-based national and regional center and argumentations why you have focused on the healthcare of patients with neurological issues.
Conclusion is so limited. There is no info about theoretical and empirical importance of these research. You should also indicate the limitations of this research and direction of future research.
Author Response
Response to Reviewer 3 Comments
The goal of this study protocol is clearly described.
The method is clear and appropriate.
Point 1: However, there is a lack of information on the Singapore-based national and regional center and argumentations why you have focused on the healthcare of patients with neurological issues..
Response 1: Please provide your response for Point 1. (in red)
We thank Reviewer 3 for her/his comments.
The study population are patients with neurological issues because some of the study team members come from the National Neuroscience Insititute. And they want to conduct a systematic study of the return on investment of interprofessional collaboration (IPC) in their setting. However, the protocol proposed by this study is also applicable to patients with other diseases cared for through IPC. As stated in the protocol paper, “If the results of our proposed study show that team-based approach produces better patient outcomes, provides care that patients need and improves HCP well-being at a lower cost, then it could serve as a persuasive example for other services within the institution to adopt similar IPC practices” (lines 169 to 172).
Point 2: Conclusion is so limited. There is no info about theoretical and empirical importance of these research. You should also indicate the limitations of this research and direction of future research.
Response 2: Please provide your response for Point 2. (in red)
The lack of definitive proof for the return of investment (ROI) of interprofessional collaboration (IPC) is likely due to two reasons: lack of a unifying conceptual framework with which to measure the healthcare outcomes of IPC, and over-reliance on the single-method study design for examining a phenome-non as complex and multi-faceted as IPC. Hence, the theoretical importance of this protocol is our suggestion for the IPC research community to use of a conceptual framework – Quadruple Aim – to guide the assessment of interprofessional collaboration’s return on investment. The empirical importance of our research is to measure IPC’s ROI by operationalizing the Quadruple Aim framework using a multimethod approach. Through this endeavor, we hope to advance the field of IPC research.
We have added a section discussing the limitations of this research and direction of future research:
This research protocol aims to address the seemingly intractable problem of measuring IPC’s ROI. Without addressing this critical gap, evidence for the ROI of IPC will remain fragmented which makes it difficult for healthcare leaders and policymakers to appreciate the value of IPC objectively. Hence, we seek to demonstrate the feasibility of measuring IPC’s ROI by operationalizing the Quadruple Aim framework using a multimethod approach. Specifically, quantitative methods are used to measure patient outcomes and cost of care, and qualitative methods to understand how team-based care impacts patient experience and provider well-being.
These aims notwithstanding, the protocol has several limitations. For study 1, the evaluation of patient outcomes adopted a single-arm design due to the unavailability of a parallel comparison group in NNI and the unavailability of electronic medical records of a historical comparison group. As a result, we are unable to make statistical comparison of the patient outcomes of team-based care with those of usual care. Instead, a non-statistical comparison with the treatment outcome for patients with the same diseases reported in the published literature will be made to interpret the findings of this part of the study. However, as this protocol is meant to test the feasibility of our concept of how to measure IPC’s ROI, future studies could prospectively plan to collect data for a comparison study.
As for Study 2, interview and observation data have inherent limitations. While semi-structured interviews may help researchers gather hard to access information such as perceptions and experiences of patients and HCP, the data may be limited by recall bias or not reflect actual interactional patterns. Observation as a mode of data collection may also raise concerns about participant reactivity, where participants may act differently because of the presence of a researcher. For these reasons, study 2 is designed to collect both interview and observation data for triangulation purposes.
Projecting ahead, if the two empirical studies described in this protocol prove to be effective in measuring the ROI of IPC, researchers may proceed to examine other important issues such as whether, how, and when healthcare budgets should be invested/disinvested in IPC. From the perspective of practical import, this will be helpful in guiding healthcare leaders and policymakers to make evidence-based decision in the provision of healthcare. From the theoretical perspective, being able to strengthen the evidence base of IPC’s ROI will also help to advance this field of research.

Round 2
Reviewer 3 Report
All comments have been included in the article.